# Challenges and Recent Advancements in COVID-19 Vaccines

**DOI:** 10.3390/microorganisms11030787

**Published:** 2023-03-18

**Authors:** Shao-Cheng Wang, Chung-I Rai, Yuan-Chuan Chen

**Affiliations:** 1Department of Psychiatric, Taoyuan General Hospital, Ministry of Health and Welfare, Taoyuan 33004, Taiwan; 2Department of Nurse-Midwifery and Women Health, National Taipei University of Nursing and Health Sciences, Taipei 112303, Taiwan; 3Health Care Business Group, Foxconn Technology Co., Ltd., New Taipei City 23680, Taiwan; 4Graduate Institute of Applied Science and Technology, National Taiwan University of Science and Technology, Taipei City 106335, Taiwan; 5Department of Nursing, Jenteh Junior College of Medicine, Nursing and Management, Miaoli County 35664, Taiwan; 6Department of Medical Technology, Jenteh Junior College of Medicine, Nursing and Management, Miaoli County 35664, Taiwan; 7Program in Comparative Biochemistry, University of California, Berkeley, CA 94720, USA

**Keywords:** SARS-CoV-2, COVID-19 vaccine, EUA, viral variant, breakthrough infection, adverse reaction

## Abstract

Vaccination is the most effective method for the prevention of COVID-19 caused by SARS-CoV-2, which is still a global epidemic. However, the evolution of SARS-CoV-2 is so rapid that various variants, including the Alpha, Beta, Gamma, Delta, and Omicron variants, have emerged, lowering the protection rate of vaccines and even resulting in breakthrough infections. Additionally, some rare but severe adverse reactions induced by COVID-19 vaccines may raise safety concerns and hinder vaccine promotion; however, clinical studies have shown that the benefits of vaccination outweigh the risks caused by adverse reactions. Current vaccines approved with emergency use authorization (EUA) were originally adaptive for adults only, and infants, children, and adolescents are not included. New-generation vaccines are needed to overcome the challenges of limited adaptive age population, breakthrough infection (mainly due to virus variant emergencies), and critical adverse reactions. Fortunately, some advances in COVID-19 vaccines have been obtained regarding enlarged adaptive populations for clinical applications, such as the Pfizer/BioNTech vaccine and the Moderna vaccine. In this article, we provide a review on the challenges and recent advancements in COVID-19 vaccines. The development of next-generation COVID-19 vaccines should lay emphasis on the expansion of adaptive age populations in all individuals, the induction of immune responses to viral variants, the avoidance or alleviation of rare but potentially critical adverse reactions, and the discovery of subunit vaccines with adjuvants encapsulated in nanoparticles.

## 1. Introduction

### 1.1. Coronavirus Structure and SARS-CoV-2 Infection

The viral infectious particle (virion) of SARS-CoV-2 has a nucleic acid core and helical symmetry of the coat protein (capsid) and the envelope. It contains about 30,000 nucleotides, which code for four structural proteins, namely the nucleocapsid (N) protein, the membrane (M) protein, the spike (S) protein, and the envelope (E) protein, as well as several non-structural proteins [1,2]. The rod-shaped spike protein is present on the envelope protein, which has the receptor binding domain (RBD) to bind with the receptor to infect the host cell [1,2]. It is activated by host-cell proteases at two distinct sites, named S1/S2 and S2′, and SARS-CoV-2 contains a cleavage site for the protease furin at S1/S2 [3]. After the RBD of the SARS-CoV-2 spike protein binds to the angiotensin-converting enzyme 2 (ACE2) receptor of human cells, the host cell initiates a mechanism for the entry of the positive-sense single-stranded RNA genome by membrane fusion [4,5,6,7,8]. Once the capsid uncoats and the RNA is released to enter the host cell, it begins to synthesize replicase and transcriptase. The replicase–transcriptase complex begins to replicate viral RNA and synthesizes its structural proteins. The involved enzymes include RNA-dependent RNA polymerase, RNA helicase, RNA 5′-triphosphatase, exoribonuclease, N7-methyltransferase, 2′-Omethyltransferase, etc., and then the replicated RNA and structural proteins assemble to form mature virions [4,7,8,9,10]. The virions are encapsulated in vesicles and delivered to the cell surface. Finally, the virions are released from the original host cells to infect new hosts through exocytosis and may cause diseases [4,7,8,9,10]. For example, the new 2019 coronavirus infectious disease, named COVID-19 by the World Health Organization (WHO), was caused by coronavirus SARS-CoV-2.

### 1.2. A Brief Historical View on COVID-19

COVID-19 spread globally from 2019~2022 and had a great impact on human lifestyle and health since its outbreak in 2019. Currently, the pandemic situation has been alleviated, and most people infected with SARS-CoV-2 may only experience mild to moderate illness and recover without any special treatment. However, some people may become seriously ill and require specific medical care or evening, including older people, infants, and those with chronic diseases (e.g., diabetes, hypertension, respiratory disease, or cancer). The best way to prevent transmission and sickness is to be well informed about how the disease spreads and to receive vaccination. Therefore, the diagnosis of SARS-CoV-2 and the development of vaccines are very crucial.

### 1.3. Major COVID-19 Vaccines around the World during 2020~2022

Vaccination is one of the most effective methods for COVID-19 epidemic prevention. Therefore, many countries have focused on developing vaccines during these years. Several COVID-19 vaccines were approved by the WHO and various countries with emergency use authorization (EUA) (Table 1) during 2020~2022. However, all vaccines were originally approved for adults only, as there were not enough data for infants, children, and adolescents. COVID-19 vaccines might result in rare but severe adverse reactions based on some clinical studies and case reports. During this period, special attention was paid to the development and health status of vaccines to ensure positive patient health. Additionally, other strategies were considered, such as the mix and match of different vaccines; the development of new vaccines, such as the mRNA vaccine; using nanoparticles as vectors; and the optimization of adjuvants to improve vaccine safety and efficacy.

### 1.4. COVID-19 Control and Prevention Is Difficult

The control and prevention of COVID-19 is very difficult because SARS-CoV-2 has the characteristics of cross-species transmission, multiple infectious routes, and high variability [4,37,38]. The viral genome of SARS-CoV is highly mutated because of some specific factors, such as RNA’s sensitivity to certain environments (e.g., temperature), RNA being easy to change or damage, and the poor proof-reading ability of RNA-dependent RNA polymerase. Many viral variants emerge all over the world easily and frequently. Thus, the development of vaccines is quite challenging. Even if vaccine development is successful, its effective period is usually short, and its efficacy is limited. Many fully vaccinated people are still infected with SARS-CoV-2, resulting in severe symptoms, or even death in some cases.

## 2. The Challenges of COVID-19 Vaccines

### 2.1. The Adaptive Age Population Is Limited

Almost all COVID-19 vaccines were originally adaptive to adults only, excluding adolescents, children, and infants, because the population involved in clinical trials are all adults (Table 1) [11,12,13,14,15,16,17,18,19,20,21,22,23,24,25,26,27,28,29,30,31,32,33,34,35,36]. The related data regarding infants, children, and adolescents are still insufficient in clinical trials. It is possible that children are less easily harmed by COVID-19; however, the side effects of vaccination may have been underestimated because the sample sizes for random clinical trials of children were too small. However, infants, children, and adolescents can also be infected with SARS-CoV-2, and the infection may more easily cause the development of moderate and even severe symptoms than in adults. 

### 2.2. The Emergency of Viral Variants

Because of the need for viral survival and propagation, the evolution of SARS-CoV-2 variants is inclined to be more infectious, highly variable, and have a lower lethal rate for host cells. This results in diverse viral variants emerging to escape vaccine prophylaxis and resist drugs, even leading to breakthrough infection.

#### 2.2.1. Breakthrough Infection

The condition for pathogens to evade the protection of vaccines is called breakthrough infection, that is, the event in which an infectious disease is still diagnosed even after vaccination [39,40,41]. The USA Center for Disease Control and Prevention (CDC) defines a breakthrough infection of COVID-19 as “diagnosed with SARS-CoV-2 at least 14 days after full vaccination against COVID-19”. Possible reasons for breakthrough infection are as follows [39,40,41]: (1) Vaccine protection is not 100%. Although many COVID-19 vaccines have obtained official approval or EUA from the WHO and various countries, the claimed protection is about 60–95% according to data from clinical trials; (2) Infection occurs prior to complete protection. Some COVID-19 vaccines (e.g., Pfizer/BNT, Moderna, AZ vaccines, etc., excluding the Johnson vaccine) require two doses to achieve full protection, and it also takes 2 weeks for antibody production to occur after vaccination. A breakthrough infection case may occur before the vaccine is completely administered or while the body is producing antibodies; (3) Differences in the genome of vaccine recipients. Vaccines fight pathogens by triggering the immune system of recipients, which is quite different from direct treatment of diseases using medicines. Therefore, the effectiveness and protection of vaccines often vary according to individual physique, such as the titer of the antibody produced and the strength of the immune response, etc. Additionally, groups with weakened immune systems may become reinfected due to insufficient immune responses or antibodies; and (4) The emergence of viral variants. Because RNA-dependent RNA polymerase of SARS-CoV-2 has poor proof-reading ability for repair during replication, the mutation rate of SARS-CoV-2 is high. Viral variants can easily escape from the protection of current vaccines.

Among the above four reasons for the occurrence of breakthrough infection, the emergence of viral variants is the major one. We speculate that some viral variants may be resistant to vaccines and decrease their effectiveness. Many vaccinated patients are still infected with SARS-CoV-2 variants, and vaccination seems to be limited in effectiveness regarding the prevention of COVID-19. However, the signs and symptoms of COVID-19 in vaccinated patients are usually moderate or. In some cases, the infection may present without any signs and symptoms; only very few cases are severe or fatal compared to unvaccinated ones. Fortunately, most vaccinated patients can easily recover from diseases, though some of them with mild, moderate, and severe symptoms may need to take drugs to relieve their symptoms. 

#### 2.2.2. Classification of Viral Variants

In March 2021, the US CDC and the World Health Organization (WHO) developed new standards for the classification of SARS-CoV-2 variants, which are divided into “Variant being monitored (VBM)”, “Variant of interest (VOI)”, “Variant of concern (VOC)”, and “Variant of high consequence (VOHC)”. These standards aim to clarify the degree of variation, risk of transmission, and seriousness of public health threat [42,43]. They are divided as follows: (1) VBM. The data indicate that variants have a potential or clear impact on approved or authorized medical countermeasures, or they are correlated with more severe diseases or increased transmission. However, they are no longer detected or only slightly circulate [42,43]; (2) VOI: Variations in the viral gene sequence that alter the ability to bind to cellular receptors and reduce the neutralizing effect, therapeutic efficacy, and diagnosis of antibodies produced after infection or vaccination. They may even increase the transmission or severity of the disease. This may result in a surge or cluster outbreak of confirmed cases, which can cause limited epidemics in multiple countries simultaneously [42,43]; (3) VOC: There is evidence that the virus is more contagious or causes more severe infections, including increased hospitalization or death. The neutralization ability of antibodies produced by previous infection or following vaccination is significantly reduced. The effectiveness of drugs or vaccines is decreased or causes a test to be in vain [42,43]; (4) VOHC. The viral variant significantly reduced the effectiveness of vaccines, drugs, and other methods. Even after vaccination, cases of breakthrough infection still surge and protection against severe diseases is very low, resulting in more patients developing severe symptoms and requiring hospitalization [42,43].

The WHO has cooperated with institutions and researchers to monitor and evaluate the evolution of SARS-CoV-2. The emergence of viral variants that cause an increased threat to global public health facilitate the characterization of specific VOIs and VOCs. The WHO and its international network of experts are monitoring variants so that countries in the world and the public can be informed about any viral changes that need a response and to prevent viral variant transmission. Global systems have been established and are being strengthened to detect “signals” of potential VOIs or VOCs [43]. 

The information and relative risk levels of formerly circulating VOCs during 2020~2022 are presented in Table 2. However, viral variants of SARS-CoV-2 have been emerging frequently and rapidly. Among these viral variants, the Omicron variant is currently the most significantly increased one in COVID-19 patients. Novel variants are transmitting globally and have spread much faster and are more contagious than other variants of SARS-CoV-2 because Omicron variants have special epidemiological and biological characteristics.

### 2.3. Rare but Potentially Critical Adverse Reactions Are Possible

Vaccinations have provided a promising strategy to control the COVID-19 pandemic, but some COVID-19 vaccines induce adverse reactions as follows: (1) very common reactions (≥1/10), including headache, diarrhea, pain at the injection site, general weakness, and induration at the injection site; (2) common reactions (≥1/100~<1/10), including dizziness, drowsiness, vomiting, muscle pain, and redness at the injection site; and (3) uncommon reactions (≥1/1000 to <1/100), including fever, itching at the vaccination site, chills, rash, nasopharyngitis, oropharyngeal pain, and palpitations. However, further studies, including specific systematic reviews and meta-analyses, have been performed to investigate other unknown side effects of COVID-19 vaccines and to explore the correlation between vaccination and adverse events during 2021~2022 [51,52,53,54,55,56]. Based on some rare and special case reports, there have been preliminary data and/or evidence to suggest that COVID-19 vaccines may play a role in causing rare but potentially critical adverse reactions (Table 3). 

The most promising and effective approach to controlling the COVID-19 epidemic is still global vaccination. However, adverse reactions post-vaccination in rare and special cases or reports may raise concerns about the safety profile of COVID-19 vaccines. Rare but potentially critical adverse reactions can be generally classified into the following four major types: hematologic diseases; neurologic diseases; cardiovascular complications (e.g., arrhythmias, myocarditis, pericarditis, venous thromboembolism); and hypersensitivity. In fact, the correlation between vaccination and critical adverse reactions are still under investigation and their relationship is indefinite. Currently, the advantages of vaccination against COVID-19 have been shown to be significantly more than the disadvantages, including mortality and morbidity caused by COVID-19, based on the benefit–risk evaluation. 

## 3. Recent Advancements in COVID-19 Vaccines

Many countries have tried to increase the rate of vaccination to reduce severe cases and control the prevalence of COVID-19. However, some factor may hinder vaccination promotion, such as the limitation of the adaptive age population, breakthrough infection mainly resulting from the emergency of viral variants, and safety concerns due to rare but potentially critical adverse reactions. Recently, significant advances in COVID-19 vaccines have been presented. Some of them have been applied clinically; for example, the adaptive age population has been expanded from adults only to include adolescents and children.

### 3.1. Expansion of Adaptive Age Population

Currently, the Pfizer/BioNTech and Moderna vaccines have provided some clinical trial data to support the notion that their adaptive application is able to expand to adolescents and children, although we may need more data on hospitalizations and deaths resulting from the adverse outcomes of COVID-19 in children (SARS-CoV-2 infections rarely cause severe symptoms or death in children). Additionally, critical phase II and III clinical trials are needed and long-term human clinical trials are required to ensure long-term human safety.

The USA FDA issue EUA for the Pfizer/BioNTech vaccines for use in individuals aged ≥16 years on 11 December 2020. As of 30 July 2021, only the Pfizer/BioNTech vaccine is authorized for adolescents aged 12–17 years, and the EUA was expanded to include adolescents aged 12–15 years on 10 May 2021 [67]. Based on the results from a phase 3 clinical trial from 14 December 2020 to 16 July 2021 in the USA, the Vaccine Adverse Event Reporting System (VAERS) received 9246 reports following Pfizer/BioNTech vaccination in this age group. They found 90.7% of them experienced mild adverse events and only 9.3% experienced serious adverse events, including myocarditis (4.3%) [67]. Approximately 129,000 adolescents aged 12–17 years were enrolled after Pfizer/BioNTech vaccination, and they reported local reactions (63.4%) and systemic reactions (48.9%) with a rate similar to that of preauthorization clinical trials for adults; systemic reactions were usual after a second dose. The US CDC and FDA continue to perform vaccine safety surveillance and provide data for vaccination guidance [67]. The results demonstrate that the Pfizer/BioNTech vaccine is probably safe for adolescents aged 12–17 years old [67]. However, the problem is that these trials only followed participants for two months after a second dose and then they crossed placebo participants over, allowing them to receive the vaccine. The enduring benefit for preventing viral infections may be lower than 90 or 95%. 

Furthermore, the USA FDA amended EUA for the Pfizer/BioNTech vaccine to expand its application to children aged 5–11 years, given in two doses (10 μg, 0.2 mL each) 3 weeks apart. As of December 19, 2021, only the Pfizer-BioNTech vaccine is authorized for administration to children aged 5–17 years [68]. In preauthorization clinical trials, the Pfizer/BioNTech vaccine was administered to 3109 children aged 5–11 years. They found most of the adverse events were mild to moderate, and no serious adverse events were reported [68]. Approximately 8.7 million doses of the Pfizer/BioNTech COVID-19 vaccine were administered to children aged 5–11 years. The VAERS received 4249 reports of adverse events after vaccination with Pfizer/BioNTech COVID-19 vaccine in this age group, of which 4149 (97.6%) cases were nonserious [68]. Approximately 42,504 children aged 5–11 years were enrolled after Pfizer/BioNTech vaccination. The data showed that a total of 17,180 (57.5%) local reactions and 12,223 systemic (40.9%) reactions, including injection-site pain, fatigue, or headache, were reported after two-dose vaccination [68]. These preliminary findings are similar to those from preauthorization clinical trials for adults. The results recommend the Pfizer/BioNTech vaccine for children aged 5–11 years, deemed safe for the prevention of COVID-19 [68].

Creech et al. assigned children (6~11 years old) in a 3:1 ratio to receive two doses of the Moderna vaccine or placebo, administered 28 days apart in an observer-blinded, placebo-controlled expansion evaluation of the selected dose in a phase 3 trial [69]. The objectives were to evaluate vaccine safety in children and the noninferiority of immune response of these children to that of young adults (18~25 years old), and to determine the incidences of confirmed SARS-CoV-2 infection. In part 1 of the trial, 751 children received 50 μg or 100 μg Moderna vaccine, and the 50 μg dose level was selected for part 2 [69]. In part 2 of the trial, 4016 children were randomly assigned to receive two doses of the Moderna vaccine or placebo and were followed for a median of 82 days after the first injection [69]. Most adverse events were low-grade and transient, and no serious adverse events were observed. One month after the second injection, the neutralizing antibody titer in children who received the Moderna vaccine at a 50 μg level was 1610, compared to 1300 at the 100 μg level in young adults [69]. The efficacy was 88.0% against SARS-CoV-2 14 days or more after the first injection when B.1.617.2 (delta) was the dominant circulating variant [66]. The results suggest that two 50 μg doses of Moderna vaccine were safe and effective in inducing immune responses and preventing children aged 6 to 11 years from contracting COVID-19 [69].

### 3.2. Induction of Immune Responses to Viral Variants

The mutation rate of SARS-CoV-2 is high; therefore, various variants have been produced, leading to most drugs and vaccines, which were formerly considered effective and useful but are now less effective or useless. The emergence of SARS-CoV-2 variants may induce breakthrough infection and threaten the controlling progress of the COVID-19 pandemic. For example, the Omicron variant escapes immune responses from post-infection or vaccination and has a greater risk of reinfection among recovered patients. Alternatively, it may cause breakthrough infection among vaccinated people. These results allow us to consider vaccinations useless and hesitate to inject vaccines. Fortunately, some improved vaccines have recently been developed to prevent SARS-CoV-2 variants.

Accorsi et al. evaluated the effectiveness of COVID-19 vaccines against the rapidly spreading SARS-CoV-2 Omicron variant to inform public health guidance [70]. This clinical trial included 23,391 cases (13,098 Omicron; 10,293 Delta) and 46,764 controls. A total of 18.6% (*n* = 2441) of Omicron cases and 6.6% (*n* = 679) of Delta cases were reported for receipt of three mRNA (Pfizer/BioNTech or Moderna) vaccine doses, as well as 39.7% (*n* = 18,587) of the controls [70]. However, these were reported for 55.3% (*n* = 7245), 44.4% (*n* = 4570), and 41.6% (*n* = 19,456), respectively, prior to receipt of two mRNA vaccine doses, and being unvaccinated was reported for 26.0% (*n* = 3412), 49.0% (*n* = 5044), and 18.6% (*n* = 8721), respectively [70]. For three doses vs. being unvaccinated, the adjusted odds ratio was 0.33 for Omicron and 0.065 for Delta. For three vaccine doses vs. two doses, the adjusted odds ratio was 0.34 for Omicron and 0.16 for Delta [68]. The median cycle threshold values were significantly higher in cases with three doses vs. two doses for both the Omicron and Delta variants [70]. Although the higher odds ratios had less protection for Omicron than for the Delta variant, the results revealed that receipt of three doses of mRNA vaccine was related to protection against both the Omicron and Delta variants, compared to being unvaccinated and receiving three doses [70]. 

Lu et al. determined the serum neutralizing antibody (nAb) titers against ancestral virus or variants in a prospective cohort study with 135 recovered COVID-19 patients [71]. They found the mean Omicron nAb titer in serum was statistically significantly higher among Pfizer/BioNTech vaccine recipients compared to non-vaccinated individuals among these recovered patients [71]. The Omicron seropositive rates in the live-virus nAb test were statistically significantly higher among Pfizer/BioNTech (90.6%) and CoronaVac (36.7%) vaccine recipients, compared with non-vaccinated individuals (12.3%) [71]. CoronaVac vaccine recipients showed that Omicron seropositive rates were higher among individuals with two doses than those with one dose (85.7% vs. 21.7%) [71]. The results suggest that boosting serum-neutralizing activity against the Omicron variant among recovered patients by Pfizer/BioNTech and CoronaVac vaccines would be an important strategy to guide vaccine policies in countries short of COVID-19 vaccines [71].

Shinde et al. randomly assigned human immunodeficiency virus (HIV)-negative adults aged 18 to 84 years or medically stable HIV-positive participants aged 18 to 64 years in a 1:1 ratio to receive two doses of either the Novavax vaccine or placebo in a phase 2a-b trial [72]. Of 6324 participants, 4387 received at least one injection of the Novavax vaccine or placebo. Approximately 30% of the participants were seropositive for SARS-CoV-2 at baseline. Among the 2684 baseline seronegative participants (94% HIV-negative and 6% HIV-positive), 15 participants in the vaccine group had mild-to-moderate COVID-19 and 29 had COVID-19 in the placebo group (vaccine efficacy, 49.4%) [72]. Vaccine efficacy among the HIV-negative participants was 60.1%. The vaccine efficacy against B.1.351 was 51.0% among the HIV-negative participants [72]. The results suggest that the Novavax vaccine is effective for the prevention of COVID-19 with higher efficacy among HIV-negative participants. Most infections were caused by the B.1.351 variant [72].

### 3.3. Avoidance or Alleviation of Rare but Potentially Critical Adverse Reactions

Generally, adverse reactions of vaccination are variable, depending on the individual’s genetics. It is difficult to make significant scientific evaluations for adverse reactions of vaccines. Severe side effects are still poorly understood due to its very low incidence rate. It is likely that exploration of the possible mechanisms of side effects induced by vaccines and medical conditions for vaccine side effects correlating to COVID-19 severity is feasible to avoid or alleviate rare but potentially critical adverse reactions [73,74]. 

Warren et al. characterized the immunologic mechanisms underlying allergic reactions to FDA-authorized mRNA COVID-19 vaccines [75]. This study included 22 patients with suspected allergic reactions to these vaccines and suspected allergy cases were identified and invited for follow-up testing. In the allergy testing, they conducted skin-prick testing to test polyethylene glycol (PEG) and polysorbate 80 (P80), and histamine and filtered saline were used as negative controls for internal validation [75]. Basophil activation was also tested after stimulation at 37 °C for 30 min. They determined the possible mechanism by analyzing the concentrations of IgG and IgE to PEG. They found that 10 of 11 cases (91%) showed positive results in the basophil activation test to PEG and 11 of 11 (100%) showed positive results in the basophil activation test to mRNA vaccines. No PEG IgE was detected in these patients, but PEG IgG was found in tested patients who were allergic to this vaccine [75]. The testing results revealed that women and those with allergic history seem to experience risky mRNA vaccine administration. IgG (non-IgE-mediated) immune responses to PEG may be associated with allergic reactions in most individuals [75]. 

Greinacher et al. identified determinants of a thromboembolic complication termed vaccine-induced immune thrombotic thrombocytopenia (VITT), induced by the AstraZeneca (ChAdOx1 nCoV-19) vaccine using biophysical techniques, mouse models, and by the analysis of VITT patient samples [76]. They found vaccine components formed antigenic complexes with platelet factor 4 (PF4) on platelet surfaces on which anti-PF4 antibodies from VITT patients are observed under a super-resolution microscope, and the PF4/vaccine complex formation was charge-driven and elevated by DNA addition. In animal studies, vaccine administration increased vascular leakage in mice, resulting in the systemic spread of vaccine components to stimulate immune responses [76]. Taking these together, PF4/vaccine complex formation and vaccine-induced proinflammation stimulated pronounced B-cell responses that caused the formation of high-avidity anti-PF4 antibodies in VITT patients [76]. The results suggest that PF4/adenovirus aggregates induced by vaccine and proinflammatory reactions trigger anti-PF4 antibody production to cause thrombosis in VITT. They show that VITT is similar to the autoimmune pathogenic mechanism of heparin-induced thrombocytopenia by two steps [76].

Hajjo et al. used an informatics strategy to study post-vaccine adverse event mechanisms by integrating analytic reports of adverse reactions from the Vaccine Adverse Event Reporting System (VAERS) using systems biology approaches [77]. They found the frequency of cardiac adverse events (e.g., myocarditis and pericarditis) was related to vaccine, vaccine type, vaccine dose, sex, and age of the vaccinated individuals [77]. Additionally, the results obtained from systems biology analysis showed that interferon-gamma (INF-γ) leads to cardiac adverse events by affecting mitogen-activated protein kinase (MAPK) and the Janus kinase/signal transducer and activator of transcription (JAK/STAT) signaling pathways [77]. They suggested that the prolongation of the time interval between vaccine doses can decrease the occurrence of inflammatory adverse reactions [77]. The results reveal that their informatic workflow can render a useful tool to study adverse events after vaccination on the systems biology level to suggest effective pharmacotherapy based on mechanisms and/or suitable preventive methods [77].

### 3.4. Discovery of Subunit Vaccine with Adjuvants Encapsulated in Nanoparticles

The gene of viral spike proteins can be cloned to specific cells to produce harmless spike subunit proteins. Following purification and extraction of spike proteins from specific cells, this subunit protein can be used as an antigen to make a recombinant vaccine by adding adjuvants and encapsulating nanoparticles. This vaccine induces immune responses once it is injected into the human body. Nanoparticles can avoid antigens being decomposed or destroyed, and they can effectively assist the antigen to penetrate the cell membrane. An adjuvant is an ingredient used to stimulate a stronger immune response in humans receiving the vaccine. Therefore, the subunit vaccine with adjuvants encapsulated in nanoparticles is considered to be both effective and safe (with the least side effects), with the potential to maximize efficacy and minimize adverse reactions.

Jacob-Dolan et al. studied the immunogenicity and protective efficacy of the GBP510 protein subunit vaccine adjuvanted with AS03, whose trade name is SKYCovione^TM^ [78]. SKYCovione™, a recombinant protein-based vaccine, is a self-assembled nanoparticle vaccine targeting the receptor binding domain (RBD) on the spike protein of SARS-CoV-2. The study demonstrated that GBP510/AS03 induced robust immune responses in rhesus macaques and protected against high-dose SARS-CoV-2 Delta challenges [78]. When challenging the Delta strain of SARS-CoV-2, the vaccinated macaques rapidly controlled the virus in bronchoalveolar lavage and nasal swabs. Binding and neutralizing antibody responses prior to challenge are associated with protection against viral replication posterior to challenge. These data are in compliance with the data of SKYCovione™ in the phase 3 clinical trial [78]. This vaccine was jointly developed by the SK Bioscience and Institute for Protein Design at the University of Washington School of Medicine, utilizing GSK’s pandemic adjuvant. This is the first COVID-19 vaccine candidate adjuvanted with GSK’s pandemic adjuvant approved by Korea on 29 June 2022.

Arunachalam et al. showed that the capacity of a subunit vaccine containing the RBD on spike proteins of SARS-CoV-2 displayed on an I53-50 protein nanoparticle scaffold (RBD-NP) could induce robust and long-term neutralizing antibody responses and protection against SARS-CoV-2 in rhesus macaques [79]. They evaluated five adjuvants as follows: Essai O/W 1849101; AS03; AS37; CpG1018-alum; and alum. RBD-NP immunization with AS03, CpG1018-alum, AS37, or alum stimulated substantial neutralizing antibody and helper T cell responses to provide protection in the pharynges, nares, and bronchoalveolar lava [79]. The neutralizing antibody response to viruses was maintained up to 180 days after vaccination with RBD-NP in AS03 (RBD-NP-AS03), relating to protection from viral infection [79]. The results significantly point out the efficacy of the adjuvanted RBD-NP vaccine in enhancing protective immunity against SARS-CoV-2 and promote the phase I/II clinical trials advances of this vaccine [79].

Walls et al. reported self-assembling protein nanoparticle immunogens, which are structure-based and designed computationally to stimulate potent and protective antibody responses against SARS-CoV-2 in mice [80]. The nanoparticle vaccines display 60 spike RBDs on SARS-CoV-2 in a highly immunogenic array. Although this vaccine has a five-fold lower dose, this vaccine elicited higher neutralizing antibody titers than the prefusion-stabilized spike by 10-fold [80]. The antibodies induced by nanoparticle vaccines can target multiple distinct epitopes, suggesting they are not easily affected by escape mutations and may minimize the risk of enhanced respiratory diseases associated with vaccines [80]. The high yield and stability of the assembled nanoparticles have resulted in this vaccine candidate being manufactured for clinical trials. These results highlight that the two-component nanoparticle platform enabled rapid generation of SARS-CoV-2 vaccines because RBD nanoparticle vaccines significantly elicit potent neutralizing antibody responses [80].

## 4. Conclusions

The evaluation results of vaccine efficacy and safety are often varied by region, country, race, gender, age, and individual genetics. For example, the same COVID-19 vaccine may demonstrate significant differences in the protective efficacy and side effects in the clinical trials of patients in different countries and/or regions. Consequently, the option and use of vaccines is very difficult and complex because we must simultaneously consider legality, efficacy, and safety when selecting a suitable vaccine.

The Omicron variant of SARS-CoV-2 has kept mutating and spreading worldwide at high rate since its emergence in late 2021. For example, the first-generation Omicron B.1.1.529 (BA.1) variant took over Delta’s dominant position within a few weeks until the emergence of the next subvariant, BA.1.1. The Omicron variant potentially became the next master for immune escape. Therefore, vaccine effectiveness against the Omicron variant infections is doubtful so some people are still hesitant to receive vaccination or even refuse to be vaccinated. Currently, most cases with Omicron variant infections are mild or asymptomatic. It is possible that the induction of immunity upon natural infection is more effective and safer than vaccination because the development of COVID-19 vaccines against Omicron variants are time-consuming and challenging.

The major advantages of specific COVID-19 vaccine products are as follows: (1) protection of uninfected people; (2) reduction in moderate and severe symptoms, as well as death; and (3) decreased spread rate and scope. The main disadvantages include the possibility of breakthrough infections and the concern of rare but critical adverse reactions. For the purpose of preventing pandemics and decreasing the rate of severe cases of COVID-19, the development of next-generation vaccines should focus on the expansion of adaptive age population in all individuals, the induction of immune responses to various viral variants, the avoidance or alleviation of rare but potentially critical adverse reactions, and the discovery of subunit vaccines with adjuvants encapsulated in nanoparticles [81] (Figure 1).

Compared with the developing process of influenza vaccines, we can learn from its lessons and make better COVID-19 vaccines in the future. The immune responses of the current influenza vaccine are concentrated on the hypervariable hemagglutinin (HA) head region. The target of new-generation influenza vaccines is the viral structure with less variability, such as the base of HA, neuroaminidase (NA), matrix protein (M), or nucleoprotein (NP). Other new technologies to improve the effectiveness of the vaccine include the use of nanoparticles to optimize the antigenicity of the target antigen, addition of novel adjuvants, increment of antigen dose, and stimulation of cell-mediated immunity. Similar to the influenza vaccine, the main antigen of current COVID-19 vaccines is the spike protein with high variability. The target of new-generation COVID-19 vaccines should focus on the viral structure with less variability, such as the envelope protein and NP. Additionally, the use of nanoparticles, addition of adjuvants, and induction of cell-mediated immunity are all potential methods to produce more effective and safer vaccines.

## Figures and Tables

**Figure 1 microorganisms-11-00787-f001:**
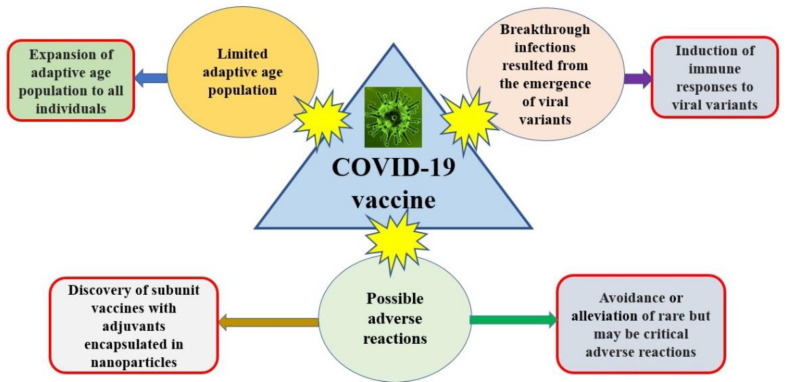
Impact of COVID-19 vaccines and development of next-generation vaccines.

**Table 1 microorganisms-11-00787-t001:** Main COVID-19 vaccines approved by the World Health Organization (WHO) or various countries during 2020~2022.

Vaccine	Manufacturer	Platform	Vector	Alleged Protection Rate	Storage	Reference
Pfizer/BioNTech	Pfizer, Inc., Brooklyn, N.Y., USA and BioNTech, Mainz, German	mRNA	Lipid nanoparticle	95% (two doses)	−70 °C	[11,12,13]
Moderna	Moderna, Inc., Cambridge, MA, USA and National Institute of Health (NIH), Bethesda, MD, USA	mRNA	Lipid nanoparticle	95% (two doses)	−20 °C	[14,15,16]
Oxford/AstraZeneca (AZ)	Oxford University, Oxford, UK and AZ plc., Cambridge, UK	DNA	Adenovirus	75% (two doses)	2–8 °C	[17,18,19]
Johnson	Johnson & Johnson Pharmaceuticals, New Brunswick, NJ, USA	DNA	Adenovirus	72% (single dose)	2–8 °C	[20,21]
BBIBP-CorV	Sinopharm Group Co., Ltd., Hong Kong, China	Inactivated vaccine	No	72% (two doses)	2–8 °C	[22,23,24]
CoronaVac	Sinovac Biotech, Beijing, China	Inactivated vaccine	No	50–80% (two doses)	2–8 °C	[25,26]
Sputnik V	Gamaleya Research Institute, Moscow, Russia	DNA	Adenovirus	90% (two doses)	2–8 °C	[27,28,29,30]
Novavax	Novavax Inc., Gaithersburg, MD, USA	Recombinant protein	Lipid nanoparticles	86% (two doses)	2–8 °C	[31,32,33]
Medigen	Medigen Vaccine Biologics Corp., Hsinchu County, Taiwan	Recombinant protein (protein subunit)	No	80~90% (two doses)	2–8 °C	[34,35,36]

**Table 2 microorganisms-11-00787-t002:** The identification, emergency, changes, and neutralizing antibody activity of SARS-CoV-2 variants, called variants of concern (VOCs), during 2020~2022.

Identification		Emergency		Changes Relative to Previously Circulating Variants		Neutralizing Antibody Activity		Date of Designation	Reference
WHO label	PANGO lineage	First outbreak country	Current circulation	Transmissibility	Mortality	From infection	From vaccination		[42,43,44]
Omicron	B.1.1.529	South Africa	Yes	Possibly increased	−63% (69–74%) related to Delta	Increased reinfection rate	Efficacy reduction against symptomatic disease, unknown for severe disease	VUM: 24 November 2021 VOC: 26 November 2021	[42,43,45,46,47,48,49,50]
Delta	B.1.617.2	India	No	+97% (76–117%)	+137% (50–230%)	Reinfections have smaller occurrence rate than vaccinated infections	Efficacy reduction for non-severe diseases	VOI: 4 April 2021 VOC: 11 May 2021 Previous VOC: 7 June 2022	[42,43,45,46,47,48,49,50]
Gamma	P.1 (B.1.1.28.1)	Brazil	No	+38% (29–48%)	+59% (44–74%)	Reduced	Retained by many	VOC: 11 January 2021 Previous VOC: 9 March 2022	[42,43,45,46,47,48,49,50]
Alpha	B.1.1.7	UK	No	+29% (24–33%)	+50% (50% CrI, 20–90%)	Minimal reduction	Minimal reduction	VOC: 18 December 2020 Previous VOC: 9 March 2022	[42,43,45,46,47,48,49,50]
Beta	B.1.351	South Africa	No	+25% (20–30%)	Possibly increased	Reduced	Efficacy reduction against symptomatic disease, retained against severe disease	VOC: 18 December 2020 Previous VOC: 9 March 2022	[42,43,45,46,47,48,49,50]

Abbreviation: variant under monitoring (VUM); variant of interest (VOI); variant of concern (VOC); World Health Organization (WHO); the Phylogenetic Assignment of Named Global Outbreak Lineages (PANGOLIN). The reported confidence or credible interval (Crl) has a low probability, so the estimated value can only be understood as possible, not certain, or likely.

**Table 3 microorganisms-11-00787-t003:** Main COVID-19 vaccines and their rarely reported adverse reactions.

Vaccine	Reported Critical Adverse Reaction	Reference
Pfizer/BioNTech	(1) Myocarditis, pericarditis, or myopericarditis (about 1–10 cases per 100,000 persons). (2) Vein thrombosis and pulmonary thromboembolism (only one case report).	[57,58,59,60]
Moderna		
Oxford/AstraZeneca (AZ)	Vaccine-induced immune thrombocytopenia and thrombosis (VITT) (incidence apparently between 1 in 125,000 and 1 in 1,000,000).	[61]
Johnson & Johnson		
Sputnik V	Panniculitis (only one case report).	[62]
BBIBP-CorV	(1) VITT (incidence apparently between 1 in 125,000 and 1 in 1,000,000). (2) Transverse myelitis.	[63,64]
CoronaVac		
Novavax	Myocarditis (only one case report).	[65]
Medigen	Multiple evanescent white-dot syndrome (MEWDS) in the eyes (only one case report).	[66]

## Data Availability

Not applicable.

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
