# Peer review of "Challenges and Recent Advancements in COVID-19 Vaccines"

_microorganisms, 2023, doi:10.3390/microorganisms11030787_

Round 1

Reviewer 1 Report

In this manuscript, the authors describe the current advances and difficulties in vaccine development of COVID-19. While I agree that the current COVID-19 vaccines are imperfect, the authors are too strong in claiming the side effects brought by these vaccines. Based on clinical evidence, the probability of severe adverse reactions is very low, and not different from the other licensed vaccines. For instance, the authors insist that COVID-19 vaccines are ineffective, as there are still vaccinees who succumb to severe COVID-19. However, their effectiveness in reducing severe COVID-19 is generally quite good, which was not reflected in their writing. I thus recommend the authors to tone down the claims on severe adverse reactions and indicate the rates of these side effects in the manuscript.

In addition, the authors did not do a comparison on the other vaccines developed for other viruses to justify the points they have made for COVID-19 vaccines. The comparison could be made so that we can learn from their lessons and make better COVID-19 vaccines in the future.

It is also important to note that the mechanistic understanding of the severe side-effects is poorly understood due to the low rates of severe side-effects. This should be emphasised, and knowledge from  vaccines against other viruses can be considered for comparison.

Finally, the authors should better structure the manuscript. The current format jumps around from virus properties, vaccines to side effects and immunogenicity. The authors could be more systematic and cover topics in a broad sense and then narrow down to the specifics.

Author Response

In this manuscript, the authors describe the current advances and difficulties in vaccine development of COVID-19. While I agree that the current COVID-19 vaccines are imperfect, the authors are too strong in claiming the side effects brought by these vaccines. Based on clinical evidence, the probability of severe adverse reactions is very low, and not different from the other licensed vaccines. For instance, the authors insist that COVID-19 vaccines are ineffective, as there are still vaccinees who succumb to severe COVID-19. However, their effectiveness in reducing severe COVID-19 is generally quite good, which was not reflected in their writing. I thus recommend the authors to tone down the claims on severe adverse reactions and indicate the rates of these side effects in the manuscript.

Ans: 1. We have revised the statement to tone down the claim on severe adverse reactions. For example, “Rare but adverse reactions are possible” has been revised to “Rare but may be critical adverse reactions”; the reports of adverse reactions post-vaccination have raised concerns about the safety profile of COVID-19 vaccines” has been revised to “the adverse reactions post-vaccination in rare and special cases or reports may raised concerns about the safety profile of COVID-19 vaccines.” (P.6 Line 197~P.7 Line 133)

  1. In the manuscript, we have emphasized “In fact, the correlation between vaccination and critical adverse reactions are still under investigation and their correlation is indefinite. Currently, the advantages of vaccination against COVID-19 have been shown to be significantly more than the disadvantages including mortality and morbidity caused by COVID-19, based on the benefit-risk evaluation.” (P.7 Line 214~224)
  2. We have indicated the rates of these side effects in Table 3. (P.6 Line 212~P.7 Line 213)

In addition, the authors did not do a comparison on the other vaccines developed for other viruses to justify the points they have made for COVID-19 vaccines. The comparison could be made so that we can learn from their lessons and make better COVID-19 vaccines in the future.

Ans: We compare the influenza vaccine with the COVID-19 vaccine to learn from its lessons and make better COVID-19 vaccines in the future. (P.12 Line 487~P.13 Line 500)

It is also important to note that the mechanistic understanding of the severe side-effects is poorly understood due to the low rates of severe side-effects. This should be emphasised, and knowledge from  vaccines against other viruses can be considered for comparison.

Ans: We have revised the statement about severe side-effects of vaccine to emphasize that it is poorly understood due to the low rates of severe side-effects. (P.9 Line 346~353)

Finally, the authors should better structure the manuscript. The current format jumps around from virus properties, vaccines to side effects and immunogenicity. The authors could be more systematic and cover topics in a broad sense and then narrow down to the specifics.

Ans: We have revised and rearranged the statement of subsection 1.1~1.3 to make the manuscript structure be more systematic and cover topics in a broad sense and then narrow down to the specifics. (P.1 Line 39~P.2 Line 87)

Reviewer 2 Report

In this work the authors reviewed newly developed COVID-19 vaccines in various platforms. The side effects of each type of COVID-19 vaccines were compared and the major challenges of current COVID-19 vaccines were discussed. The manuscript is well-written. Overall, this review is very important and timely. I would recommend it for publication in Microorganisms given the following points addressed.

Major points:

1.     At page 3, the sentence "The mutation rate of the viral genome is high because the proof-reading ability of RNA-dependent RNA polymerase is poor and the RNA molecule is not stable." is not perfectly correct for SARS-CoV-2.

2.     To add more updated information, authors might need to add or discuss about the COVID-19 vaccine with the structure based computationally designed platform, such as Skycovione. (Ref #1: The next generation of coronavirus vaccines: a graphical guide. Nature, 2023 Feb;614(7946):22-25.  doi: 10.1038/d41586-023-00220-z.

Ref #2: Adjuvanting a subunit COVID-19 vaccine to induce protective immunity. Nature, 2021 Jun;594(7862):253-258.  doi: 10.1038/s41586-021-03530-2. Epub 2021 Apr 19.

Ref #3: Elicitation of Potent Neutralizing Antibody Responses by Designed Protein Nanoparticle Vaccines for SARS-CoV-2. Cell. 2020 Nov 25;183(5):1367-1382.e17.  doi: 10.1016/j.cell.2020.10.043.  Epub 2020 Oct 31.)

Author Response

In this work the authors reviewed newly developed COVID-19 vaccines in various platforms. The side effects of each type of COVID-19 vaccines were compared and the major challenges of current COVID-19 vaccines were discussed. The manuscript is well-written. Overall, this review is very important and timely. I would recommend it for publication in Microorganisms given the following points addressed.

Major points:

  1. At page 3, the sentence "The mutation rate of the viral genome is high because the proof-reading ability of RNA-dependent RNA polymerase is poor and the RNA molecule is not stable." is not perfectly correct for SARS-CoV-2.

Ans: We have revised this sentence into “The viral genome of SARS-CoV is highly mutate because of some specific factors such as RNA is sensitive to enviroment (e.g., temperature), RNA is easy to change or even damage and the proof-reading ability of RNA-dependent RNA polymerase is poor”. (P.3 Line 92~101)

  1. To add more updated information, authors might need to add or discuss about the COVID-19 vaccine with the structure based computationally designed platform, such as Skycovione.

Ref #1: The next generation of coronavirus vaccines: a graphical guide. Nature, 2023 Feb;614(7946):22-25.  doi: 10.1038/d41586-023-00220-z.

Ref #2: Adjuvanting a subunit COVID-19 vaccine to induce protective immunity. Nature, 2021 Jun;594(7862):253-258.  doi: 10.1038/s41586-021-03530-2. Epub 2021 Apr 19.

Ref #3: Elicitation of Potent Neutralizing Antibody Responses by Designed Protein Nanoparticle Vaccines for SARS-CoV-2. Cell. 2020 Nov 25;183(5):1367-1382.e17.  doi: 10.1016/j.cell.2020.10.043.  Epub 2020 Oct 31.)

Ans: We have added a subsection 3.4 to discuss the COVID-19 vaccine with the structure based computationally designed platform such as Skycovione, and three references which the revierwer suggested are all included. (P. 10 Line 400~P.11 Line )

Round 2

Reviewer 1 Report

Revised manuscript now looks better. No further comments from me